# Anatomical View of the Internal Carotid Artery Occlusion in Japanese Black Cattle

**DOI:** 10.3390/ani14030365

**Published:** 2024-01-23

**Authors:** Arvendi Rachma Jadi, Hinako Fujisaki, Amany Ramah, Mahmoud Baakhtari, Shoichiro Imatake, Shoichi Wakitani, Masahiro Yasuda

**Affiliations:** 1Graduate School of Medicine and Veterinary Medicine, University of Miyazaki, 1-1 Gakuen Kibanadai-nishi, Miyazaki 889-2192, Japan; arvendi_rachma_jadi@med.miyazaki-u.ac.jp (A.R.J.);; 2Department of Anatomy, Faculty of Veterinary Medicine, Universitas Gadjah Mada, Fauna Street Karangmalang, Yogyakarta 55281, Indonesia; 3Laboratory of Veterinary Anatomy, Faculty of Agriculture, University of Miyazaki, 1-1 Gakuen Kibanadai-nishi, Miyazaki 889-2192, Japan; 4Department of Forensic Medicine and Toxicology, Faculty of Veterinary Medicine, Benha University, Qalyubia 13518, Egypt; 5Faculty of Veterinary Medicine, Balkh University, Balkh 1702, Afghanistan

**Keywords:** anatomy, blood supply, internal carotid artery, Japanese black cattle, occlusion

## Abstract

**Simple Summary:**

Japanese black cattle are a well-known Wagyu cattle breed in Japan, with the best-producing carcass. Based on previous studies and the literature, the existence of the internal carotid artery in cattle still needs to be determined. Thus, this research examined the arterial supply in the brains of Japanese black cattle, especially the presence of the internal carotid artery and its closure at different ages.

**Abstract:**

The internal carotid artery (ICA) is a branch of the common carotid artery (CCA), along with the external carotid artery (ECA), which together provide the blood supply for the brain. The description of the ICA in cattle is vague, including denial of its existence or degeneration at an early stage after birth. This anatomical study investigated the internal carotid artery in Japanese black cattle. Sixty-five heads of Japanese black cattle aged from newborn to 13 years were dissected and injected with colored latex from the CCA after separating the head and body. Diameter measurements of the artery branches from the CCA on its bifurcation were conducted. Furthermore, a histological examination of the ICA wall’s structures, which consist of the tunica intima, tunica media, and tunica externa, was performed. The ICA of Japanese black cattle is closed on the left side after age 3 years, except for a small lumen at 13 years, whereas the right ICA remains open at all ages. The location of occlusion of the left internal carotid artery (LICA) shows thickness of the tunica intima and an increased connective tissue layer area. The diameter of the ICA does not differ between the left and right sides, and there is no correlation with age. Therefore, further studies are needed, especially of ICA occlusion related to Japanese black cattle’s physiology or cerebrospinal disease.

## 1. Introduction

The internal carotid artery (ICA) is one of the arteries supplying the brain alongside the occipital, vertebral, and (internal) maxillary artery (MA) [1]. It branches from the common carotid artery (CCA) together with the external carotid artery (ECA), which supplies blood to the neck and head area [1,2]. The ICA passes through the foramen jugulare to enter the cranial cavity and is characterized, like in other mammals, by a loop formed [3]. Compared to the ECA, the diameter of the ICA is smaller, as is that of the occipital artery (OA) that arises close to the ICA [4]. In humans, the ICA is classified into intradural near the dural ring, intracavernous, and the petrous part covered by pericranial tissue. This muscular artery has well-developed smooth muscle in the tunica media layer and is well known for lacking the external elastic membrane [5].

In ruminants, the distal section of the ICA is a single artery that originates from a few branches of the maxillary artery that enter the cranium through the foramen ovale and foramen orbitorotundum, a complicated mesh on the floor of the cranium [6,7]. The ICA ascends along the bulla tympanica (BT) and runs in an S shape between the temporal bone’s petrosal section and the auditory tube’s cartilage component, connecting to the rete mirabile (RM) [8,9]. The RM, which splits into a rostral and a caudal division, is used to reconstruct a distal portion of the ICA [10]. The ICA has the upper rostral epidural RM and the continuous caudal epidural RM. The rostral epidural RM is in the cavernous sinus as an anastomose network, which is situated intracranially [1]. It is supported by the branches to the MA as it continues from the CCA, which becomes the middle meningeal artery and the external ophthalmic artery. Those arteries branch into the rostral rete branch (RRB) and caudal rete branch (CRB), respectively [11]. In addition, the vertebral, condylar, and occipital arteries on the caudal side comprise the caudal epidural RM, which is composed of many interlacing arterioles [12]. Moreover, the RM is commonly found in artiodactyls, including cattle, as an arterial support for supplying blood to the brain and acts as a flow-facilitating system, replacing the function of the ICA [13].

As a part of the suborder Ruminantia, cattle have an ICA that can atrophy after birth [5], diminish, then be converted into connective tissue during adulthood [1] or their first month [14], and even be absent [15]. Further information about the ICA and its occlusion in Japanese black cattle must be obtained. However, the description and clarification of the ICA and its blood supply in Japanese black cattle must be investigated via gross anatomical comparisons and described based on the findings. As such, histological observations were performed on both the left and right sides of the site of the closure of the ICA. The diameter was compared between the arteries by age and all artery branches.

## 2. Materials and Methods

### 2.1. Sample and Animal Ethics

Sixty-five heads of Japanese black cattle, ranging in age from newborn to 13 years, of different sex types (male, 19; female, 35; and steer, 11), were used for this research (Appendix A). All animal procedures were conducted with the approval of the Institutional Animal Care and Use Committee of the University of Miyazaki, Miyazaki, Japan (2021-006-01). All experiments were performed at the Laboratory of Veterinary Anatomy, Department of Veterinary Sciences, Faculty of Agriculture, University of Miyazaki, Japan.

### 2.2. Gross Anatomical Procedure and Statistical Analysis

The Japanese black cattle’s head was separated within the body and injected manually with 0.9% saline solution through the CCA (both sides) to clean the vessel of blood using a needle and syringe. Colored latex (Showa Denko Chloroprene type 842A, Showa Denko Co., Ltd., Tokyo, Japan) was injected into the CCA (approximately 60–100 mL on each side, depending on the head size), which was left in a freezer to harden the latex for 24 h afterward. Moreover, dissection was performed to identify the arteries, especially the ICA, by removing the muscle tissue, nerves, and unwanted organs. The observation was performed following a gross anatomical procedure. Photographs were then taken with a digital camera.

The diameter of the arteries connected to the ICA was measured at the bifurcations of the artery branches. The measured arteries were the CCA, ECA, ICA, OA, CRB, and RRB. After the arteries were exposed, their outer diameters were measured using a digital caliper (BLD-200, Niigata Seiki, Niigata, Japan). The data were collected and then analyzed. The analysis was conducted using the Wilcoxon signed-rank test for the left–right differences and Spearman’s test for the age and diameter of the artery. The diameters of the arteries were then compared with the Wilcoxon signed-rank test, the ICA to the OA, and the CRB–RRB, respectively (with the OA, under two years old, 57 of 65 cattle; CRB–RRB, under two years old, 28 of 65 cattle; without sex differentiation).

### 2.3. Histological Observation

Samples were collected from the cattle older than one year of age (14 of 65). Histological staining was conducted on 6 samples out of 14 after the latex passed through for young cattle. The arteries were removed from the head and then fixed in a 20% buffered formalin solution (for more than 7 days) using the micro technique with graded alcohol and toluene. The sample was then embedded in paraffin, cut to a thickness of 4 µm, and stained with hematoxylin–eosin. The ICA samples were cut continuously to the site of occlusion, or the narrowest lumen was exposed. The lumen area was divided into lumen and small lumen based on their presence. Screening for the occlusion area was performed using a light microscope. Furthermore, histological photographs were acquired with a Nano Zoomer 2.0-RS Scanner (C10730-13, Hamamatsu Photonics K.K., Shizuoka, Japan). All anatomical structures were identified based on the Veterinary Anatomical Nomenclature by The International Committee on Veterinary Gross Anatomical Nomenclature [16].

## 3. Results

### 3.1. Gross Anatomical Observations

The ICA was present in the cranium of all cattle observed in this study, as shown in Figure 1. However, the ICA branched off from the CCA at the dorsal part of the stylohyoid bone, then ascended along the BT and passed through the foramen lacerum (FL). After passing through the FL, the artery ran in an S shape between the auditory tube’s cartilage and the temporal bone’s petrosal part, connected to the RM. The ICA arises with the OA from the CCA, while the ECA has a different direction to the rostral side. The ECA branches into the CRB and RRB, which supply the RM via FL. The RM consists of the anastomosed small artery and divides into the rostral epidural RM (R) and the caudal epidural RM (C) (Figure 1d). The diameter of the ECA is more than twice that of the ICA. On the other hand, the diameter of the OA, which branches from the ECA, is slightly wider than that of the ICA (Figure 2b–d).

### 3.2. Correlation of Arterial Diameter with Age

There was no difference in the artery’s diameter between the left and right sides. In the CCA, there was a significant positive correlation between artery diameter and age in days, as shown in Figure 2a (LCCA: r = 0.604, *p* < 0.05; RCCA: r = 0.673, *p* < 0.05). The OA only had a slight positive correlation on the left side (LOA: r = 0.311, *p* < 0.05) since the right side was weaker than that (ROA: r = 0.211, *p* = 0.11) (Figure 2b). No correlation between the diameter and age was found for the ICA (LICA: r = 0.085, *p* = 0.53; RICA: r = −0.023, *p* = 0.86). In contrast, the ECA had a significant positive correlation (LECA and RECA, r = 0.711, and r = 0741, *p* < 0.05, respectively) (Figure 2c,d). No correlation was found for the CRB and RRB with age in days (LCRB: r = 0.327, *p* = 0.028; RCRB: r = 0.172, *p* = 0.38; LRRB: r = 0.125, *p* = 0.53; RRRB: r = −0.1, *p* = 0.61) (Figure 2e,f).

### 3.3. Comparison of the Diameter of the Arteries

Differences in artery diameter between the ICA and the OA or CRB–RRB were observed (57 of 65 and 28 of 65 cattle, respectively). There was a significant difference between ICA and OA diameters for both the left and right sides (*p* < 0.05) (Figure 3a). Furthermore, the RICA showed a significant difference with the right CRB–RRB (both with the right caudal rete branch (RCRB) and the right rostral rete branch (RRRB) (*p* < 0.01) (Figure 3b). However, there was a significant difference between the LICA and the left rostral rete branch (LRRB) but not the left caudal rete branch (LCRB) (*p* < 0.01) (Figure 3b).

### 3.4. Histological Structure of the Internal Carotid Artery

The tunica intima, tunica media, and tunica externa layers were observed from the collected samples via histological examination. Endothelial cells and a continuous internal elastic membrane were seen in the tunica intima. Furthermore, smooth muscle and elastic fibers were found in the tunica media, typical for a muscular-type artery. The tunica externa is formed by the connective tissues as the outermost parts of the artery. External elastic fibers were not found in the tunica externa. For example, during latex injection, the ICA of a 5-year-old Japanese black cattle was blocked. It was confirmed that the occlusion resulted from an enlargement of the tunica intima, consisting of smooth muscle and lumen area, which was not seen as a route for blood flow in the ICA (Figure 4a–e).

Based on the histological findings, Japanese black cattle aged 3 years showed a wide lumen, with residual latex inside the artery on both sides (Figure 5a). The tunica media has a thin layer compared to the tunica externa. Furthermore, the LICA of cattle 5 and 6 years of age was perfectly closed, whereas, at age 13 years, it had a small lumen area, as shown in Figure 5b–d. Compared to those, the RICA at all ages showed no occlusion. Latex was observed at 3 years; the others showed a small lumen. The RICA showed thickening of the tunica media, vacuolation of the tunica media, and obscured smooth muscle. The borderline between the tunica intima and media of the LICA, through which the latex did not pass, was indistinct. A thickened tunica intima layer was shown, with smooth muscle cells and no external elastic membrane.

## 4. Discussion

In cattle, the ICA was thought to be nonexistent [17,18] or to close early in life [14,17], and it may degenerate or vanish during embryonic development [2]. The presence of vascular casts was one of the factors that contributes to the early closure of the ICA, as documented in previous investigations [3,10,13,19,20]. In the Cervidae family, the initial fragment of the ICA was found at about 2.5 years old, proving that the ICA was obliterated at the extracranial segment. Before that age, the ICA was present and fully preserved [21]. However, the present study showed that the ICA in Japanese black cattle is present at all ages, but latex injection did not pass through the artery at some ages. The ICA was closed, particularly on the left side (LICA), at ages 5, 6, and 12 years. Despite that, the right side (RICA) remained open. Both the left and right ICAs were open at three years of age, with an evident lumen, as shown by latex injection. The ICA was still open in 13-year-old Japanese black cattle, as indicated by the open canal on both sides.

Wible described how ruminant ICA closure is caused by the movement of the temporal bone near the BT during development [22]. The variation in breed, in this case Japanese black cattle, may have resulted in a change in temporal bone displacement during development, leading to the distinct time of ICA closure. To connect to the RM, the ICA must pass through the FL and squeeze between the auditory canal’s cartilage section and the temporal bone’s petrosal region [19]. However, the obliteration of the ICA, unrelated to hearing in ruminants, may be associated with the eardrum portion’s developmental changes [21]. This process is not apparent in vessels connected to the RM. The ICA has the portion’s narrowest diameter; the blood flow to the RM of the ICA decreases throughout central development, and then the blood supply to the RM increases [19]. These facts imply that the ICA’s blood flow to the RM is lower than that of other vessels, such as the OA, CRB, and RRB, which are linked to it. This condition could cause a difference in artery diameter between the rostral and caudal rete branches and the ICA with the OA. The RM is also found in pigs and sheep [10,23]. Compared with mouse deer, which are non-dominant ruminants that lack an RM, the ICA provides the brain’s blood supply [6]. Brain blood flow may come from the RM or the ICA. The RM’s structure is believed to have contributed to the ICA’s disappearance. The ICA in sheep is not well developed, so its function is substituted with one of the ECA branches, the internal MA, involved in the circle of Willis circulation [2]. In the yak, the OA, external ophthalmic artery, condylar artery, vertebral artery, and MA, together with the ICA, supply the blood for the RM [13]. In addition, it was confirmed that, in domestic cattle, the RM is divided into the rostral and caudal epidural RM, which is unpaired for the caudal part [24].

The investigation of artery diameter demonstrated no side-to-side difference between left- and right-side arteries. As shown in Figure 2, only the CCA, ECA, and slightly the OA (on the left side) showed a significant positive correlation between artery diameter and increasing age (all *p* < 0.05). Furthermore, Ocal et al. discovered that the ICA had the thinnest diameter in the foramen jugulare area [19]. Despite that, the comparison of the ICA with the OA (*p* < 0.05, Figure 3a) and CRB–RRB showed a significant difference on both sides except on the left side for the ICA and LCRB (*p* < 0.01, Figure 3b). This finding shows that side and age may affect the artery’s diameter in some parts, but no information on sex differentiation is available. In humans, the diameter of the normal carotid artery may vary based on sex, side, and age [25]. This information may support the development of reference values for examinations conducted for surgical therapy related to cerebrovascular problems in Japanese black cattle in applications related to ICA occlusion, such as for the prevention of epistaxis in horses [26]. As a theoretical suggestion, further research is needed to investigate the reason for ICA occlusion, mainly on the left side.

Thus, the ICA still forms even if the RM has formed, and there is no atrophy but occlusion of the ICA on the left side because there is no difference in their diameters. Thus, the RM is an essential source of blood supply and storage in the brain of Japanese black cattle since the ICA closes on mostly the left side and is even almost closed on the right side (6-year-old Japanese black cattle). The ICA is responsible for carrying blood, along with the ECA and OA, which supply blood to the brain by the RM, despite the RM acting as blood storage in the brain and an efficient thermoregulator for brain temperature and protection from thermal stress or overheating by the transfer of heat to the cooler arterial blood [27,28]. The RM in artiodactyls (including cattle) is a unique vascular system supplying the brain, which confers an ability to regulate brain temperature independent of body temperature. Limitations in food, water supplies, and temperature stress caused by hot and cold environments may lead to adaptation and evolution from thermal damage [6,7,21,29]. The ICA degenerates or vanishes during embryonic development, so branches of the maxillary and ascending pharyngeal arteries enter the complex arterial system by the RM instead of the brain blood supply via the ICA [5].

The histological findings of ICA occlusion in the present investigation were comparable to those of ductus arteriosus closure. Vascular lumen closure, tunica intima thickening, endothelial cell thinning or loss, an obscured internal elastic membrane, and smooth muscle fibers were all present in the closed LICA. A tissue layer is present among endothelial cells and the internal elastic membrane (Figure 4). In the sheep muscular artery, a thick layer called the subendothelium is present between the internal elastic membrane and the endothelium [30]. On the other hand, the tunica intima is thickened, and the smooth muscle fibers are obscured in the RICA at that exact location. The internal elastic membrane is obscured, the number of elastic fibers in the tunica media increases, and endothelial cell loss occurs as the tunica intima thickens, mucoid forms, and cytolytic necrosis occurs in the tunica media [31,32,33]. The smooth muscle in the tunica media runs longitudinally, as seen in sheep [30]. On both sides, the ICA lacks the external elastic membrane in the tunica externa in the present study. Meanwhile, the external elastic membrane is lacking in humans, and the fiber layer is as robust as the internal elastic membrane. At the horizontal part of the cavernous sinus, the external elastic membrane of the ICA was found to have vanished [5].

Although it is present, the ICA of Japanese black cattle probably closes with age, mainly on the left side (Figure 5). The region close to the FL in the ICA of Japanese black cattle might have a unique structure not present elsewhere. It is believed to be where the loss of the external elastic membrane causes changes in the elasticity of the vessel wall. No external elastic membrane was discovered in any region of the internal carotid artery in the cattle examined in the present study. The ICA may regress during maturation because of the absence of the external elastic membrane on its vessel [5]. The differentiation between the right and left ICA relates to the lack of lumen area, which causes the right side to keep the blood flow running on the head. Since occlusion of the ICA in Japanese black cattle occurs on the left side, it may lead to an anatomical species variation as age passes or possibly through the evolution processes of the head’s blood circulation.

In humans, occlusion of the ICA may cause some cerebrovascular diseases, such as carotid stenosis and atherosclerotic plaques, leading to ischemic stroke [24,34]. Anatomically, the human ICA is divided into segments as described by anatomical location classifications [35,36]. In contrast, the ICA in animals still needs to be updated into specific segments, as in humans. Further study of ICA differentiation in animals is needed to determine where occlusion occurs. On the other hand, ICA occlusion in humans is one of the pivotal causes of cerebrovascular disease, which mainly increases the future risk of strokes [36]. Therefore, it is essential to systematically study ICA occlusion related to Japanese black cattle’s physiology or cerebrovascular disease.

## 5. Conclusions

This study found that Japanese black cattle’s internal carotid artery (ICA) remains open from newborn until the age of three years. Then, its lumen becomes smaller until it closes, especially on the left internal carotid artery (LICA). The occlusion of the ICA in Japanese black cattle is thought to be caused by the presence of the rete mirabile (RM) and its complex pathways, displacement of the temporal bone at the time of development, and the histological structure of the ICA. The blood supply in Japanese black cattle may be sufficient through the RM, so the ICA may have been occluded as part of the evolution in Japanese black cattle. Further research on this finding should examine the occlusion and its possible relationship with physiological anomalies or vascular diseases.

## Figures and Tables

**Figure 1 animals-14-00365-f001:**
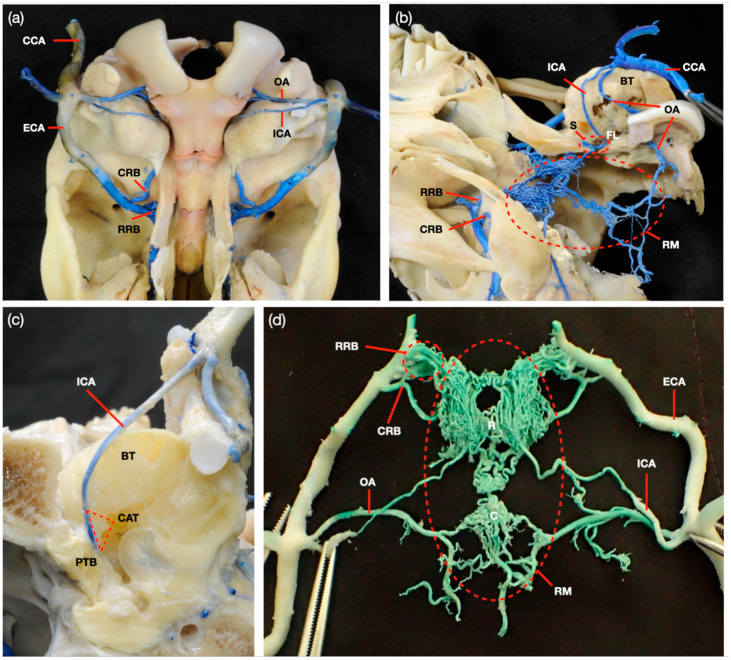
Arterial vascularization of the Japanese black cattle’s cranium. (**a**) Ventral view, (**b**) ventrolateral view, (**c**) the ICA runs between the cartilage of the auditory tube (CAT) and the petrosal part of the temporal bone (PTB), and (**d**) ventral view of the rostral epidural rete mirabile. Common carotid artery (CCA), occipital artery (OA), internal carotid artery (ICA), external carotid artery (ECA), caudal rete branch (CRB), rostral rete branch (RRB), bulla tympanica (BT), S shape (S), foramen lacerum (FL), rete mirabile (RM), rostral epidural rete mirabile (R), and caudal epidural rete mirabile (C).

**Figure 2 animals-14-00365-f002:**
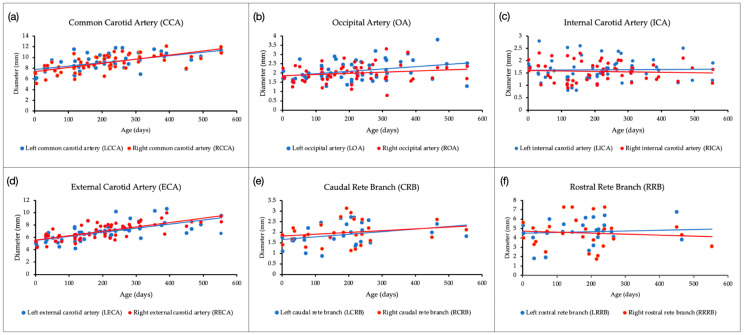
Scatter plot graphs of the correlation between the diameter of cranium arterial vascularization (left–right) and age (in days) of Japanese black cattle (57 of 65 cattle for (**a**–**d**); 28 of 65 cattle for (**e**,**f**)). (**a**) Common carotid artery (CCA), (**b**) occipital artery (OA), (**c**) internal carotid artery (ICA), (**d**) external carotid artery (ECA), (**e**) caudal rete branch (CRB), and (**f**) rostral rete branch (RRB). The red and blue dots show the right and left arteries, respectively.

**Figure 3 animals-14-00365-f003:**
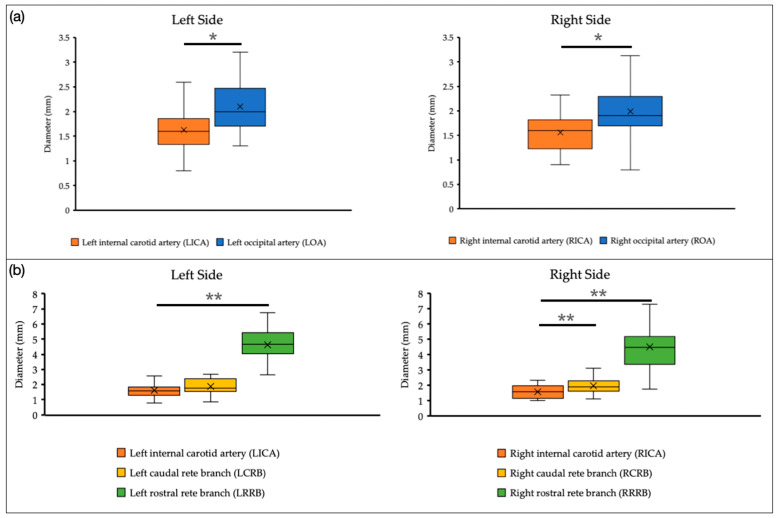
Box plot graphs of diameters comparing the internal carotid artery (ICA) with the occipital artery (OA) and CRB–RRB on the left and right sides (* = *p* < 0.05, ** = *p* < 0.01). (**a**) The internal carotid artery (ICA) with the occipital artery (OA) (57 of 65 cattle). (**b**) The internal carotid artery (ICA) with the CRB–RRB (28 of 65 cattle).

**Figure 4 animals-14-00365-f004:**
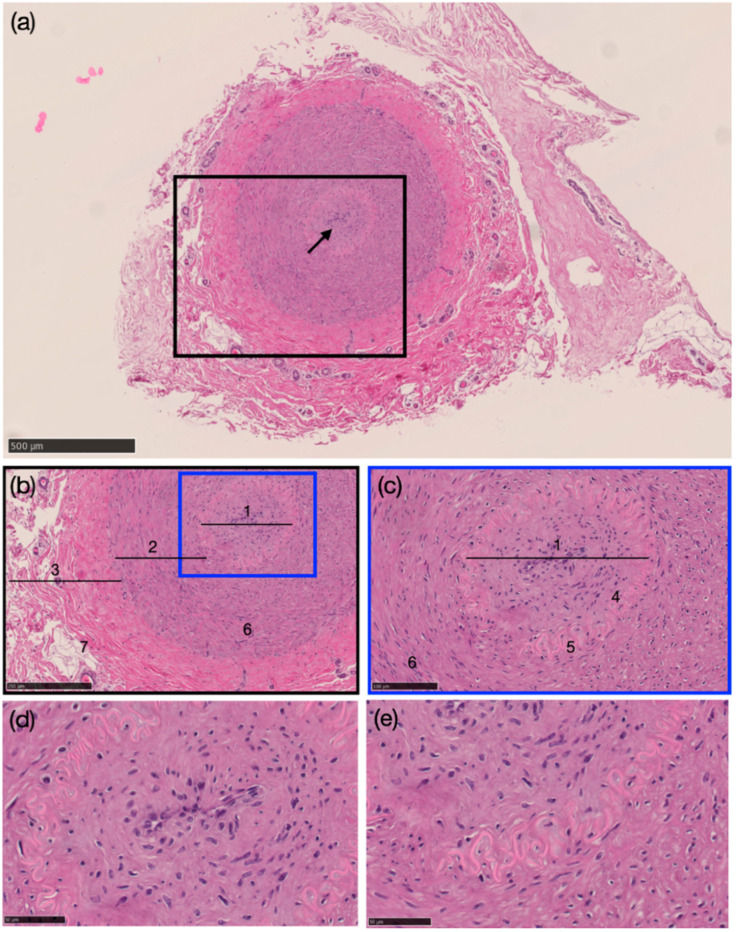
The morphology of the left internal carotid artery (LICA) of a 5-year-old Japanese black cow (hematoxylin–eosin staining). (**a**) Occlusion of the internal carotid artery with no lumen (arrow) (black scale bar = 500 μm), (**b**) closer view of the LICA (black scale bar = 250 μm), (**c**) closer view of the tunica intima and tunica media area (black scale bar = 100 μm), (**d**) higher magnification of area number 1 (black scale bar = 50 μm), and (**e**) higher magnification of area number 5 (black scale bar = 50 μm). 1. Tunica intima, 2. tunica media, 3. tunica externa, 4. endothelial cells, 5. internal elastic membrane, 6. smooth muscle fiber, 7. fibroelastic connective tissue.

**Figure 5 animals-14-00365-f005:**
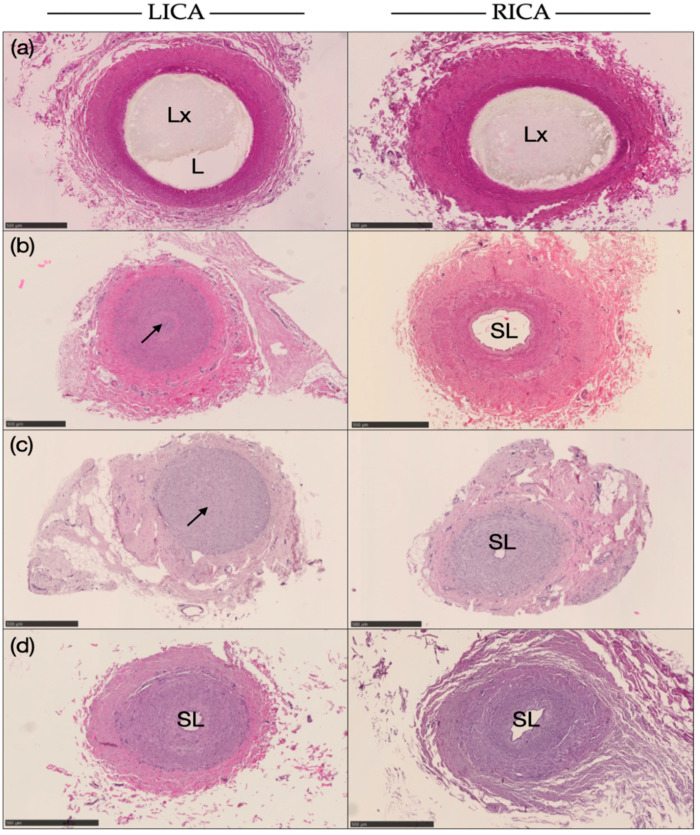
Histological comparison of the left carotid artery (LICA) and the right carotid artery (RICA) at several ages in Japanese black cattle. Some arteries have lumina, whereas others show no lumen, meaning they are completely closed (especially on the left side): (**a**) 3 years old, (**b**) 5 years old, (**c**) 6 years old, and (**d**) 13 years old (black scale bar = 500 μm). Arterial lumen (L), injected latex (Lx), small lumen artery (SL), and site of occlusion (shown with arrow).

## Data Availability

Further information and data on this study are available to the corresponding author upon request.

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
