# Peer review of "Anatomical View of the Internal Carotid Artery Occlusion in Japanese Black Cattle"

_animals, 2024, doi:10.3390/ani14030365_

Round 1

Reviewer 1 Report

Comments and Suggestions for Authors

Dear Authors, 

Your paper is interesting, pointing to a classical yet not fully tackled issue regarding the internal carotid artery's morphology in large ruminants. 

When I saw the description, I just remembered to check for one of the most important to the al reference sources of classic veterinary anatomy- at least in Europe- the “bible” of Barone...and I saw it it not cited. There you also can find some things regarding the development and disappearance/involution of the ICA. I suggest you should cite Barone as well along Konig and the classical sources. 

On the other hand, let me give you my pointed observations: 

  1. Clarify the origin of the ICA in ruminants. Refer to NAV for OA and EC arteries origins in introduction 

  1. Line 58- alisphenoid canal? Can you please clarify? (nomenclature Understood that you try a parallel with carnivores, but is it really ncesessary?) 

  1. Lines 73-75 needs rephrasing. A bit awquard 

  1. In methodology part you should mention exactly the method of injection for the latex and the possible difference in case of manual injection  

  1. Results section- I feel that a raw metrical data table should be inserted, as tables and histograms might not be providing precise figures for another researcher to compare with (but take it as an advice if you want to keep the raw data) 

  1. Lines 138-144- I would insist a little more on the correlation data as 

  1. Lines 148-149..not clear for me..can you please explain a little this difference?57/65 vs 28/65? 

  1. Lines 151-152 as raw data is missing .... questionable 

  1. Lines 165-166...why 14/65? 

  1. Line 173..can you be more specific? 

  1. Figure 3..n=????? 

  1. Scatter and bocx plots very ok...but in the absence of raw data they seem little problematic- see line 5 as well 

  1. Lines 221-233..the discussion on RM seems redundant, please see which section you should insert it..either here or into the introductory part- same for lines 270-288 

  1. Discussions- why reference to carnivores? Maybe you should stick to ruminants only in this section... 

  1. Lines 312-314...good observation, but can you at least try an explanation for R vs L situation? I do not expect you to clarify it now, but an attempt is more than welcomed...this would be a beautiful closure for your paper that otherwise leaves the same question unanswered as it is in the summary and introductory part

Author Response

Response Review Report Form 1

Dear reviewer,

Thank you very much for the insightful comment and suggestion as great feedback for improving our manuscript. We are grateful and honored to be reviewed by experts in Veterinary Anatomy. As a consideration, we added and cited Getty-Sisson and Grossman as a substitution with Barone for the classical veterinary anatomy literature because of the limitation of Barone Books (English version). The answers we provide are in the order of your comment list. The revised texts from the manuscript are shown in red.

Furthermore, these are our responses (in red) based on your comments:

  1. Clarify the origin of the ICA in ruminants. Refer to NAV for OA and EC arteries origins in the introduction 

Based on NAV, it is explained that ICA is listed upper the RM epidurale rostrale which is supplied by the Maxillary artery, and continuous as RM epidurale caudale. The occipital artery also mentioned as artery vertebralis in other animals which is along the lateral surface of the Axis and Atlas, is represented only by small branches that join the Ramus descendens and Ramus anastomoticus cum a. occipitali.

That paragraph seems unclear, so we added some sentences based on Getty-Sisson and Grossman’s Book to make the explanation better provided in lines 43-47 (added as a new reference).

Lines 43–47:

The internal carotid artery (ICA) is one of the arteries supplying the brain alongside the occipital, vertebral, and (internal) maxillary artery (MA) [1]. It branches from the common carotid artery (CCA) together with the external carotid artery (ECA), which supplies blood to the neck and head area [1,2].

  1. Line 58- alisphenoid canal? Can you please clarify? (nomenclature Understood that you try a parallel with carnivores, but is it really necessary?) 

The reason for that is that we want to show that ICA in other animals, such as dogs, foxes, and cats (decreased and led to obstruction by the bone structure). Furthermore, after your great recommendation, we deleted it and stuck to Ruminant only. We deleted some sentences based on the reference numbers 7 to 11 (which will changed into a new reference list order). We also provide some sentences as seen in lines 55-63.

Lines 55-63:

The ICA ascends along the bulla tympanica (BT) and runs in an S-shape between the temporal bone’s petrosal section and the auditory tube’s cartilage component, connecting to the rete mirabile (RM) [8,9]. The RM, which splits into a rostral and a caudal division, is used to reconstruct a distal portion of the ICA [10]. The ICA has listed the upper rostral epidural RM and continuous as caudal epidural RM. The rostral epidural RM is in the cavernous sinus as an anastomose network, which is situated intracranially [1]. It is supported by the branches to the MA as it continues from the CCA, which becomes the middle meningeal artery and the external ophthalmic artery.

  1. Lines 73-75 needs rephrasing. A bit awquard 

Thank you for the comment, the updated sentence is described in lines 70-73.

Lines 70-73:

As a part of the suborder Ruminantia, cattle have ICA that can atrophy after birth [5], diminish, then be converted into connective tissue during adulthood [1] or their first month [14], and even absent [15]. Further information about ICA and its occlusion in Japanese black cattle must be boarded.

  1. In methodology part you should mention exactly the method of injection for the latex and the possible difference in case of manual injection  

Thank you for the suggestion. Our method is manual injection through the CCA on the cervix area. After finding the CCA (big artery around the neck), needle was inserted into the CCA (both sides) and tightened with the cotton thread. After that, clear the blood with the 0.9% saline solution until the saline solution flows out of the vessel (neck area). The colored latex was injected through the CCA using a syringe 20ml by the same needle. Depending on the head size, it is approximately 3-5 syringes (60-100ml) for the whole head on both sides. Lastly, please keep it in the freezer to harden the latex.

We revised as seen in lines 89-95:

The Japanese black cattle’s head was separated within the body and injected manually with 0.9% saline solution through the CCA (both sides) to clean the vessel of blood using the needle and syringe. Colored latex (Showa Denko Chloroprene type 842A, Showa Denko Co., Ltd., Tokyo, Japan) was injected into the CCA (approximately 60-100 ml on each side depending on the head size), which was left in the freezer to harden the latex for 24 hours afterward.

  1. Results section- I feel that a raw metrical data table should be inserted, as tables and histograms might not be providing precise figures for another researcher to compare with (but take it as an advice if you want to keep the raw data) 

Thank you for the suggestion about the raw metrical data. We provide the raw data as the supplementary data. Please check the supplementary data for further details.

  1. Lines 138-144- I would insist a little more on the correlation data as 

Thank you for the comment but maybe your comment is not finished yet. If we understand based on your comment (from lines 138-147), the correlation data is provided in Figure 2 (2a to 2f). We added some sentences and the correlation data in paragraph 3.2 (listed below and in red in the manuscript).

Lines 138-147:

There was no difference in the artery’s diameter between the left and right sides. In the CCA, there was a significant positive correlation between artery diameter and age in days, as shown in Figure 2a (LCCA: r=0.604, p<0.05; RCCA: r=0.673, p<0.05). The OA only had a slight positive correlation on the left side (LOA: r=0.311, p<0.05) since the right side was weaker than that (ROA: r=0.211, p=0.11)(Figure 2b). No correlation between the diameter and age was found for the ICA (LICA: r=0.085, p=0.53; RICA: r=-0.023, p=0.86). In contrast, the ECA had a significant positive correlation (LECA and RECA, r=0.711, and r=0741, p<0.05, respectively) (Figure 2c and 2d). No correlation was found for rami retis (CRB and RRB) with age in days (LCRB: r=0.327, p=0.028; RCRB: r=0.172, p=0.38; LRRB: r=0.125, p=0.53; RRRB: r=-0.1, p=0.61) (Figure 2e and 2f).

  1. Lines 148-149..not clear for me..can you please explain a little this difference?57/65 vs 28/65? 

The number of samples is:

(a) The internal carotid artery (ICA) with the occipital artery (OA) (the number of the samples is 57 of 65 cattle), n=57.

The total sample that we used was 65 cattle, but unfortunately, it is only 57 of the OA samples can be analyzed. Please also check the supplementary data that we provided.

(b) The internal carotid artery (ICA) with the CRB-RRB (the number of the samples is 28 of 65 cattle), n=28.

The total sample that we used was 65 cattle, but unfortunately, not all samples performed the measurements because of difficulties during exposure to the CRB-RRB. So, it is only 28 samples that can be performed for the measurements. Please also check the supplementary data that we provided.

  1. Lines 151-152 as raw data is missing .... questionable 

We provide the raw data as supplementary data. Please check the supplementary data for further details.

  1. Lines 165-166...why 14/65? 

The reason behind the sample collection is that only 14 of 65 samples are related to the age of the cattle. The calves under two years old mostly show the latex through the artery during the latex injection. Due to that reason, we picked the cattle three years to 13 years old from the histological investigation (six samples).

  1. Line 173..can you be more specific? 

Thank you for the comment, the updated sentence is described in lines 173-175. This sentence still relates to Figure 4 which showed the morphology of LICA of five years old Japanese black cattle.

Lines 173-175:

It was confirmed that occlusion resulted from an enlargement of the tunica intima, consisting of smooth muscle and lumen area, which was not seen as a route for blood flow in the ICA (Figure 4a-e).

  1. Figure 3..n=????? 

Figure 3 depicts the diameters comparison of the internal carotid artery (ICA) with the occipital artery (OA) and rami retis (CRB and RRB) on the left and right sides (p<0.05). The number of samples is:

(a) The internal carotid artery (ICA) with the occipital artery (OA) (the number of the samples is 57 of 65 cattle), n=57.

The total sample that we used was 65 cattle, but unfortunately, it is only 57 of the OA samples can be analyzed. Please also check the supplementary data that we provided.

(b) The internal carotid artery (ICA) with the CRB-RRB (the number of the samples is 28 of 65 cattle), n=28.

The total sample that we used was 65 cattle, but unfortunately, not all samples performed the measurements because of difficulties during exposure to the CRB-RRB. So, it is only 28 samples that can be performed for the measurements. Please also check the supplementary data that we provided.

  1. Scatter and bocx plots very ok...but in the absence of raw data they seem little problematic- see line 5 as well 

We appreciate your compliment about using the Scatter and Plots Boxes. We will provide the raw data as supplementary data. We apologize, we do not understand which one is line 5, maybe there is a mistake number. Please check the supplementary data that we provided.

  1. Lines 221-233..the discussion on RM seems redundant, please see which section you should insert it..either here or into the introductory part- same for lines 270-288 

Thank you for the comment about redundant sentences that explaining about RM. We decided to add some of the sentences to the introduction part. We moved the sentences into lines 55-63 in the introduction, whereas for discussion changed into lines 270-277.

Lines 55-63:

The ICA ascends along the bulla tympanica (BT) and runs in an S-shape between the temporal bone’s petrosal section and the auditory tube’s cartilage component, connecting to the rete mirabile (RM) [8,9]. The RM, which splits into a rostral and a caudal division, is used to reconstruct a distal portion of the ICA [10]. The ICA has listed the upper rostral epidural RM and continuous as caudal epidural RM. The rostral epidural RM is in the cavernous sinus as an anastomose network, which is situated intracranially [1]. It is supported by the branches to the MA as it continues from the CCA, which becomes the middle meningeal artery and the external ophthalmic artery.          

Lines 270-277:

The RM in artiodactyls (including cattle) is a unique vascular system supplying the brain, which confers an ability to regulate brain temperature independently of body temperature. Limitations of food, water supplies, and temperature stress caused by hot and cold environments may lead to adaptation and evolution from thermal damage [6-7,23,31]. The ICA degenerates or vanishes during embryonic development, so branches of the maxillary and ascending pharyngeal arteries enter the complex arterial system by the RM instead of the brain blood supply via the ICA [5].

  1. Discussions- why reference to carnivores? Maybe you should stick to ruminants only in this section... 

At first, we just wanted to add comparative information related to the ICA in other animals, but we decided to stick to the ruminants only. Thank you for the insightful suggestion. We revised the discussion part line 239, and we deleted previous references (which will changed into a new reference list order).

Line 239: The RM is also found in pigs and sheep [21,24].

  1. Lines 312-314...good observation, but can you at least try an explanation for R vs L situation? I do not expect you to clarify it now, but an attempt is more than welcomed...this would be a beautiful closure for your paper that otherwise leaves the same question unanswered as it is in the summary and introductory part.

Thank you for your suggestion. We revised the sentences on lines 299-305 based on your comment.

Lines 299-305:

The ICA may regress during maturation because of the absence of the external elastic membrane on its vessel [5]. The differentiation between the right and left ICA relates to the lack of the lumen area, which causes the right side to keep the blood flow running on the head. Since occlusion of the ICA in Japanese black cattle occurs on the left side, it may lead to an anatomical species variation as age passes or possibly by the evolution processes of the head’s blood circulation.

Reviewer 2 Report

Comments and Suggestions for Authors

In the article "Anatomical Views of The Internal Carotid Artery Occlusion in Japanese Black Cattle" the Authors describe findings on the process of obliteration of the internal carotid artery in Japanese Black Cattle. The work is interesting, but the results in my opinion are too confident. To support the results, Authors need to broaden the research and add new methodology. Please consider rethinking and rewriting the results.

Line 22 - the existence of the internal carotid artery in cattle is debatable - the existence of this vessel in cattle is not debatable according to the newest literature

Line 43 - the ICA itself does not supply the eye, in most mammalian species its a. ophthalmica externa that branch off the a. maxillaris - please specify the species

Line 45-46 - ICA is the vessel entering the carotid canal in cattle, it divides from the CCA before the canal

Line 61-63 - circle of Willis is not the same as rete mirabile. Circle of Willis is well-described structure among different species of animals and it is created by the arteries of the baseof the brain that supply the encephalon, and the rete mirabile is the structure from which intracranial part of the ICA emerges

Line 64 - the main vessel supplying the rostral rete mirabile are the rostral branches to the rostral epidural rete mirabile (mostly from the maxillary artery -also may branch off of ophthalmica externa)

2.1 - please provide information on age distribution of all individuals used in the study - it is very important for the results, especially if you found ICA in every specimen

The article should also have much more photographs of individual specimens showing the ICA in different ages - it can be included as a supplementary material

It is important to give serious evidence on ICA existance in older animals. In Cervidae the ICA obliterates later than 600 days old (showed on your analysis - why?) - "The internal carotid artery in the ontogenesis ofselected representatives of the Cervidae family in Poland" by Zdun, Butkiewicz and FrÄ…ckowiak

The existance of small lumen in a vessel does not provide an evidence of functionality of the vessel as the blood supplier. This results needs more advanced imaging to be completely true - angio CT or fluoroscopy may be useful in such study. I think the results should be reconsidered and formed less confident. 

Line 166 - why only 16 samples were collected for histological examination? What what the age distribution among those samples?

169-170 - HE staining is not suitable for elastin fibers examination. Please consider adding trichrome staining samples or high quality magnification of the wall of the vessels 

Please add the measurements of the vessels lumen. The fact that the vessel is not completely obliterated does not equal its a functioning vessel.

Author Response

Response Review Report Form 2

Dear Reviewer,

Thank you for your time, effort, and insightful feedback for improving our manuscript. On behalf of our team, we are honored to be reviewed by experts in Veterinary Anatomy. The answers we provide are in the order of your comment list. We revised the result and discussion part according to your suggestion. We also added some information about the methodology. The revised texts from the manuscript are shown in red (the line numbering follows the new manuscript).

Furthermore, these are our responses (in red) based on your comments:

  1. Line 22 - the existence of the internal carotid artery in cattle is debatable - the existence of this vessel in cattle is not debatable according to the newest literature.

Thank you for the comments. Regarding your comment, we provide a new sentence related to that in lines 21-22.

            Lines 21-22:

Based on the previous studies and the literature, the information about the internal carotid artery in cattle still needs to be determined and boarded.

  1. Line 43 - the ICA itself does not supply the eye, in most mammalian species its a. ophthalmica externa that branch off the a. maxillaris - please specify the species

Thank you very much for your correction. We added a new sentence based on your comment and the reference in lines 43-48.

Lines 43-48:

The internal carotid artery (ICA) is one of the arteries supplying the brain alongside the occipital, vertebral, and (internal) maxillary artery (MA) [1]. It branches from the common carotid artery (CCA) together with the external carotid artery (ECA), which supplies blood to the neck and head area [1,2].

  1. Line 45-46 - ICA is the vessel entering the carotid canal in cattle, it divides from the CCA before the canal.

Because of previous sentence refers to a reference from the sea turtle, we decided to delete it and provide a new sentence based on your comment and checking the reference as shown in lines 46-48 (based on reference in ruminant).

Lines 46-48:

The ICA passes through the foramen jugulare to enter the cranial cavity and is characterized, like in other mammals, with a loop formed [3].

  1. Line 61-63 - circle of Willis is not the same as rete mirabile. Circle of Willis is well-described structure among different species of animals and it is created by the arteries of the base of the brain that supply the encephalon, and the rete mirabile is the structure from which intracranial part of the ICA emerges

We focused on the RM despite of Circle of Willis. We provide a new sentence based on your comment and checking the reference as shown in lines 55-63.

Lines 55-63:

The ICA ascends along the bulla tympanica (BT) and runs in an S-shape between the temporal bone’s petrosal section and the auditory tube’s cartilage component, connecting to the rete mirabile (RM) [8,9]. The RM, which splits into a rostral and a caudal division, is used to reconstruct a distal portion of the ICA [10]. The ICA has listed the upper rostral epidural RM and continuous as caudal epidural RM. The rostral epidural RM is in the cavernous sinus as an anastomose network, which is situated intracranially [1]. It is supported by the branches to the MA as it continues from the CCA, which becomes the middle meningeal artery and the external ophthalmic artery.

  1. Line 64 - the main vessel supplying the rostral rete mirabile are the rostral branches to the rostral epidural rete mirabile (mostly from the maxillary artery -also may branch off of ophthalmica externa)

We provide a new sentence based on your comment and checking the reference as shown in lines 60-63.

Lines 60-63:

The rostral epidural RM is in the cavernous sinus as an anastomose network, which is situated intracranially [1]. It is supported by the branches to the MA as it continues from the CCA, which becomes the middle meningeal artery and the external ophthalmic artery.

  1. 1 - please provide information on age distribution of all individuals used in the study - it is very important for the results, especially if you found ICA in every specimen

Thank you for the comments, we provided supplementary data for the information about the age distribution (in days). Please check the supplementary data.

  1. The article should also have much more photographs of individual specimens showing the ICA in different ages - it can be included as a supplementary material

For this paper, we provide only some figures from our findings. We provided supplementary data for the information. We found the occlusion part in the ICA from Japanese black cattle aged 5, 6, and 12 (not shown in Figure 5) on the left side, whereas the right side is still open with a wide lumen and or smaller lumen. Therefore, cattle aged 10 and 13 found no occlusion on both sides, with a small lumen area. Unfortunately, our 10-year sample decided not to be shown due to not being occluded perfectly, and only one collected on one side. Sample three years old had no occlusion since the latex passed through the vessel. It is also experienced by all the young calves samples under two years old.

  1. It is important to give serious evidence on ICA existance in older animals. In Cervidae the ICA obliterates later than 600 days old (showed on your analysis - why?) - "The internal carotid artery in the ontogenesis of selected representatives of the Cervidae family in Poland" by Zdun, Butkiewicz and FrÄ…ckowiak

For this paper, we conducted the histological examination of 14 from 65 Japanese black cattle based on their age. The samples are above one year old until 13 years old. The young calves showed the latex passing through the artery during the latex injection. Due to that reason, we picked the cattle above one until 13 years old from the histological investigation. For the age distribution, please check the supplementary data that we provided.

Thank you for the paper suggestion. We also appreciate Mr. Maciej Zdun and the team, who are concerned about the vascular circulation in the heads of some species. We also cited some of his papers related to our research on Japanese black cattle. We provide some sentences based on the reference that you suggested in lines 217-219 and 231-233.

Lines 217-219:

In the Cervidae family, the initial fragment of the ICA was found at about 2,5 years old as a prove that ICA obliterated at the extracranial segment. Before that age, the ICA was present and fully preserved [23].

Lines 231-233:

However, the obliteration of the ICA, unrelated to the hearing in ruminants, may be associated with the eardrum portion's development changes [23].

  1. The existance of small lumen in a vessel does not provide an evidence of functionality of the vessel as the blood supplier. This results needs more advanced imaging to be completely true - angio CT or fluoroscopy may be useful in such study. I think the results should be reconsidered and formed less confident. 

In this research, we are not claiming that a small lumen of the ICA makes the vessel dysfunctional but still as a way for blood circulation. If the occlusion occurs in some cattle without any specific symptom, the blood flow in Japanese black cattle heads should not affect the cattle condition. We investigated the existence of the occlusion in ICA and the possible reason behind that. However, ICA in cattle is not the biggest blood supplier for the brain compared to the MA, which is larger in size than ICA.

Thank you for your recommendation about using the angio CT or fluoroscopy. Unfortunately, our challenge during this research was the limitation of time for the sample collection since the samples needed to be executed as soon as possible.

  1. Line 166 - why only 16 samples were collected for histological examination? What the age distribution among those samples?

For this paper, we conducted histological examination for 14 of 65 Japanese black cattle samples. The reason behind the samples is related to the age of the cattle. The calves under two years old show that the latex passed through the artery during the latex injection. Due to that reason, we do not perform histological examinations for young calves. We picked cattle three to 13 years old for the histological investigation (six samples). For the age distribution, please check the supplementary data that we provided.

  1. 169-170 - HE staining is not suitable for elastin fibers examination. Please consider adding trichrome staining samples or high quality magnification of the wall of the vessels 

Thank you for the recommendation. Since we did not perform the Trichrome staining, we added new magnification for Figure 4. We provide (d) and (e) for Figure 4.

  1. Please add the measurements of the vessel’s lumen. The fact that the vessel is not completely obliterated does not equal it’s a functioning vessel.

Thank you for the suggestion. We did not provide the measurement of the vessel lumen because we do not focus on that. This research focused on the occlusion in ICA in Japanese black cattle. We need to do further research to confirm the lumen diameter, but since we did the diameter comparison between the left and right sides of the artery, it may give a better perspective on that. 

Round 2

Reviewer 2 Report

Comments and Suggestions for Authors

I want to thank the Authors on responding to my letter. The changes made increased the value of the article. This is ready for publication in its present form. Congratulations!